

# Expression status and clinical significance of lncRNA *APPAT* in the progression of atherosclerosis

Fanming Meng[1], Jie Yan[2], Qiongshan Ma[1], Yunjuan Jiao[1], Luyang Han[1], Jing Xu[3], Fan Yang[4] and Junwen Liu[1]

[1] Department of Histology and Embryology, School of Basic Medical Sciences, Central South University, Changsha, Hunan Province, People's Republic of China
[2] Department of Forensic Science, School of Basic Medical Sciences, Central South University, Changsha, Hunan Province, People's Republic of China
[3] Department of Otolaryngology-Head and Neck Surgery, Xiangya Hospital, Central South University, Changsha, Hunan Province, People's Republic of China
[4] Department of Internal Neurology, Tongliao KEQU First Hospital, Tongliao, People's Republic of China

Corresponding author
Junwen Liu, liujunwen@csu.edu.cn

## ABSTRACT

**Background.** Long non-coding RNAs (lncRNAs) have been reported to modulate cardiovascular diseases, and expression dynamics of lncRNAs in the bloodstream were proposed to be potential biomarkers for clinical diagnosis. However, few cardiovascular diseases-related circulating lncRNAs were identified and their prediction power has not been investigated in depth. Here we report a new circulating lncRNA, *atherosclerotic plaque pathogenesis associated transcript* (*APPAT*), and evaluated its role and predicting ability in atherosclerotic development.

**Methods.** *APPAT* was analyzed and screened by high-throughput sequencing, and then detected *in vitro* and *in vivo*. Immunofluorescence-fluorescence in situ hybridization (IF-FISH) was utilized to explore distribution and subcellular location of *APPAT*. The expressing alteration of *APPAT* in samples of healthy and pathological coronary artery was explored further. We also assessed the level of circulating *APPAT* in blood samples from healthy individuals, and patients with angina pectoris (AP) or myocardial infarction (MI). Additionally, we predicted and validated microRNA targets of *APPAT*, then showed the expression level of a candidate target which was primarily measured in human VSMCs cell line, coronary artery, and blood samples. Lastly, we examined the potential indicating ability of *APPAT* for the risk of AP or MI.

**Results.** *APPAT* showed significant reduction in ox-LDL treated human VSMCs *in vitro*. It enriched in contractile VSMCs of artery tunica media and mainly existed in cytoplasm. Significant down-regulation of *APPAT* was found in coronary artery samples with severe stenosis. More importantly, we observed decreased expression of *APPAT* in blood samples accompanying disease progression. ROC and correlation analyses further verified the relatively high predicting ability of *APPAT*. We also observed the predicted miRNA exhibited opposite expression direction to that of *APPAT*.

**Conclusions.** This study revealed that circulating lncRNA-*APPAT* may perform an important function and have some indicating ability on the development of atherosclerosis.

## INTRODUCTION

Atherosclerosis concerns a chronic inflammatory progress which is caused by pathological vascular change. The process of atherosclerotic plaques accumulating on the arterial wall results in many cardiovascular problems like angina pectoris (AP), coronary disease, myocardial infarction (MI) and heart failure (*Harada et al., 2014*; *Libby, Ridker & Hansson, 2011*). Dysfunction or activation of multiple cell types in the artery wall under stress of stimuli risk factors are involved in the pathogenesis of atherosclerosis (*Shan et al., 2016*; *Zhou et al., 2016*). Activated vascular smooth muscle cells (VSMCs) play a vital role in the progression and stability of atherosclerotic plaques (*Gorenne et al., 2013*; *Lovren et al., 2012*; *Wang et al., 2015*). Lesions accumulating on artery wall, the alteration of VSMCs phenotype, extracellular matrix deposition, and formation of fibrous cap will all contribute to the thickening and stiffening of arterial wall (*Toba et al., 2014*).

LncRNA is generally defined as transcripts that are longer than 200 nucleotides and absent in potential of coding proteins. Mechanistic models are starting to explore their functions under both physiological and pathological conditions (*Leone & Santoro, 2016*). A few studies have investigated that dynamic expression and multiple functions of lncRNA closely associate with the progress of atherosclerosis through regulating the function of endothelial cell, phenotype of VSMCs, metabolism and inflammation, although they were suggested to be transcriptional noises before (*Bell et al., 2014*; *Bochenek et al., 2013*; *Ellis, Graham & Molloy, 2014*). Increasing evidences have proved that ncRNAs exist in bloodstream, which suggest the potential of circulating lncRNAs in serving as diagnostic biomarkers by testing their dynamic expression in body fluids.

Stabilized secondary structures of lncRNA enable their detection in blood or urine as free nucleic acids (*Reis & Verjovski-Almeida, 2012*). Such a non-invasive test in blood would enable early detection of at-risk individuals. Research of circulating lncRNA mostly focused on their role in cancer development monitoring or classification so far (*Liang et al., 2016*; *Su et al., 2015*; *Zhang et al., 2016*). However, the secretion, transportation and stabilization of circulating lncRNAs, and the precise molecular mechanisms of how lncRNAs function remains unclear. To date, research about circulating lncRNA in pathogenesis of cardiovascular disease is just at the beginning (*Cai et al., 2016*; *Kumarswamy et al., 2014*; *Yan et al., 2016*; *Yang et al., 2015*).

Here we uncovered a circulating lncRNA ENST00000620272, renamed as *atherosclerotic plaque pathogenesis associated transcript* (*APPAT*). *APPAT* was screened by high throughput sequencing analyses, possibly acting as a contractile VSMCs cytoplasmic lncRNA in coronary artery tissue. We further found *APPAT* was stable in plasma and its dynamical expression could be validated in patients with atherosclerotic disease. The *APPAT* exhibited a clinical significance and might be contribute to the non-invasive diagnosis of atherosclerosis progress.

## MATERIALS AND METHODS

### Cell culture and ox-LDL treatment

Rabbit and human VSMCs (Cellbio, Shanghai, China) were cultured in RPMI 1,640 culture medium (Invitrogen, Carlsbad, CA, USA) with 10% fetal calf serum, 100 units/ml penicillin and 100 μg/ml streptomycin. The ox-LDL was used as stimuli on cultured rabbit and human VSMCs. Cells were exposed to ox-LDL (Yiyuan Bio, Guangzhou, China) in concentration of 80 μg/ml for 48 h, and cells without ox-LDL treatment were left as control group.

### RNA-Seq: preparation, sequencing, genome mapping, and transcriptome assembly

Total RNA of each sample was isolated using TRIzol reagent (Invitrogen, Carlsbad, CA, USA). The NanoPhotometer spectrophotometer (IMPLEN, Westlake Village, CA, USA), the Qubit RNA Assay Kit in Qubit 2.0 Fluorometer (Life Technologies, Carlsbad, CA, USA), and the RNA Nano 6000 Assay Kit of Bioanalyzer 2100 System (Agilent Technologies, Santa Clara, CA, USA) ware utilized to check the purity, concentration, and integrity of RNA respectively. Eight cDNA libraries in total were constructed from ox-LDL-treated rabbit VSMCs ($n = 4$) and control group ($n = 4$). Sequencing libraries were generated using NEB-Next Ultra Directional RNA Library Prep Kit for Illumina (NEB, Ipswich, MA, USA) followed by library fragments purification and quality assessment. Finally libraries were sequenced on Illumina HiSeq 4,000 Platform, and paired-end reads with 150 bp were generated by Illumina HiSeq 2,500 Platform. We screened out clean reads of high-quality from raw data (66.1 GB, SRA accession ID: SRP124805) for subsequent analyses. The rabbit (*Oryctolagus cuniculus*) genome (OryCun2.0 in the NCBI Assembly database) was used as reference genome for reads mapping through TopHat v2.0.9 (*Kim et al., 2013*; *Trapnell, Pachter & Salzberg, 2009*). Cufflinks v2.1.1 was used for mapped reads assembly of each sample (*Trapnell et al., 2010*).

### LncRNA identification and expression

All successfully assembled reads were combined by Cuffcompare software. Then qualified transcripts (≥200 bp, or ≥2 exons, ≥3 reads coverage) were screened out. Comparison between transcripts and reference rabbit lncRNAs was performed by Cuffcompare software. RNA transcripts like tRNA, rRNA, snoRNA, snRNA, pre-miRNA, and pseudogenes were also detected and discarded. Software CNCI, CPC, PFAM, and phyloCSF were used to assessing the coding potential of transcripts (*Kong et al., 2007*; *Lin, Jungreis & Kellis, 2011*; *Mistry, Bateman & Finn, 2007*; *Sun et al., 2013*).

The expression levels of lncRNAs in each sample were evaluated using Cuffdiff (v2.1.1). Fragments per kilo-base of exon per million fragments (FPKM) for expressing evaluation followed the previous method (*Trapnell et al., 2010*). The threshold of "$P < 0.05$, FC (fold change) >2 or <0.5, and False discovery rate (FDR) <0.05" was used to judge the significance of gene expression differences between control and ox-LDL-treated group. FDR was generated by Cuffdiff.

## Participants and ethical approval

Blood samples of clinical patients were collected and classified as normal subjects ($n = 43$, ages: 54–72), AP ($n = 48$, ages: 53–79) and MI ($n = 47$, ages: 48–75) after clinical diagnose from The Third Xiangya Hospital of Central South University. MI was diagnosed through combination of several clinical parameters: ischemic symptom plus increased cardiac troponin I (cTnI) and creatine kinase-MB (CK-MB), pathological Q wave.

The studies have been performed in accordance with the ethical standards laid down in the 1964 Declaration of Helsinki. The Institutional Ethics Committee (The Third Xiang Ya Hospital of Central South University) approved the study and written informed consent was obtained from all patients (approval number: 2015-S177).

## Isolation of human plasma

For lncRNA detection, whole blood samples (5 ml per patient) were collected from subjects via a direct venous puncture into tubes containing sodium citrate, centrifuged at 3,000 rpm for 15 min, then the supernatant (plasma) was carefully transferred into an RNase-free tube and restored in $-80\ °C$.

## Obtain and classification of coronary artery samples

Coronary artery samples were gained from forensic pathological cadaver autopsy of Center of Forensic Service of Xiangya of Hunan Province. The cadavers were all selected and judged by forensic pathological identification, then classified as normal ($n = 21$) and coronary artery stenosis ($n = 28$) based on carefully checking under microscope.

## Total RNA extraction, amplification and quantitative real-time PCR (qPCR)

Total RNAs of rabbit and human VSMCs were isolated using Eastep$^{TM}$ Total RNA Extraction Kit (Promega, Beijing, China). Blood samples were extracted by RNeasySerum/Plasma Kit (Qiagen, Hilden, Germany), and RNeasy FFPE Kit (Qiagen, Hilden, Germany) for FFPE total RNA. Amplification of lncRNA and miRNAs was performed by GoScript$^{TM}$ Reverse Transcription System (Promega, Madison, WI, USA) and Bulge-Loop$^{TM}$ miRNA qRT-PCR Primer Kit (RiboBio, Guangzhou, China) respectively. The qPCR was performed on Applied Biosystems$^{®}$ 7500 Real-Time PCR System with SYBR green method (Applied Biosystems, Foster City, CA, USA). Glyceraldehyde phosphate dehydrogenase (GAPDH) was used as endogenous reference for lncRNA expression while U6 for miRNAs expression.

## LncRNA immunofluorescence and fluorescence *in situ* hybridization (IF-FISH)

Slides were processed by combined IF-FISH protocol which developed for the simultaneous detection of protein and RNA. The slides were hybridized by Cy3-probes specific for the RNA. Subsequently, IF was performed on the same slides. The α-SMA for contractile VSMCs were purchased from Abcam (Cambridge, MA, USA). Incubated the slides with anti-protein for 24 h at 4 °C. The reaction was developed with FITC Affinipure Goat Anti-Mouse IgG (Sigma, St. Louis, MO, USA). Nuclei were counterstained with DAPI (Beyotime,

Shanghai, China). IF-FISH analysis was performed by epifluorescent microscope (Leica DMI6000B).

### Prediction and primary validation of miRNA targets for *APPAT*

The pictar, miRDB, lncBase and mirna22 online databases were used for predicting miRNAs targets of *APPAT*. The intersection of these predicting results assemblages was calculated. Then candidate miRNAs were primarily selected as which were at least included in three databases. Further screening work was based on an integrated consideration of hybrid ability, number of biding site, sequence identity, and research background of them. The expressing change of miRNAs in human VSMCs was detected after ox-LDL treatment.

### Statistical analysis

Statistical significance was assumed at $P < 0.05$. And statistical significance of the means ($\pm$standard deviation, SD) which were determined by Student's $t$-test or one way ANOVA analyses. False discovery rate (FDR) was utilized for multiple testing with a threshold of FDR <0.05 (*Van den Oord, 2008*). Receiver operating characteristic (ROC) curves were calculated and the area under the curve (AUC) were evaluated subsequently. Pearson's correlation analyses was employed to verify the relationship between *APPAT* and screened miRNA in blood sample. Graphpad prism 5 (GraphPad Software Inc., La Jolla, CA, USA) was used to perform analyses and graphs plot. Medcalc (v17.9.7) was used for AUCs comparison.

## RESULTS

### Identification of differential expressed lncRNAs in ox-LDL treated rabbit VSMCs

The high-throughput RNA-seq technique was used to analyze the expression of lncRNAs in ox-LDL treated VSMCs of rabbit. A total of 42,533,007 clean reads were generated after discarded those reads with poly-N >10%, adapters, or any other type of contaminants. All clean reads were mapped to the reference genome of rabbit, and the final mapping rate was ranged from 71.39 to 74.23% in all rabbit VSMCs samples. The Cufflinks results indicated that 147,153 transcripts were assembled at the first step.

Then sequences were qualified and the rest transcripts were blasted with latest transcriptome of rabbit lncRNAs as well as known types of RNAs. Three thousand, six hundred thirty presumed lncRNAs transcripts were detected and all of these lncRNAs were recognized as novel sequence without any report before. After protein-coding predicting analyses, 2,037 novel lncRNAs were identified with no protein-coding ability for following analysis (Fig. 1A).

The expression level of lncRNA transcripts were estimated, and a total of 28 lncRNA transcripts differentially expressed in ox-LDL treated rabbit VSMCs group comparing to control group, including 17 up-regulated transcripts and 11 down-regulated (FDR <0.05, $P < 0.05$). These 28 differentially expressed transcripts corresponded to 28 lncRNA unknown rabbit genes. Among them, seven transcripts were significantly up-regulated as well as five down-regulated (Fold change $\geq$2, FDR <0.05, $P < 0.01$) (Table S1).
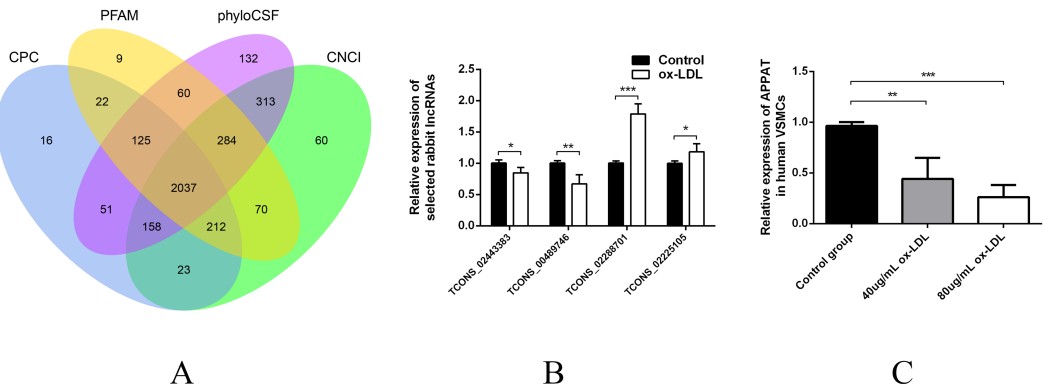

**Figure 1** **Validation of high throughput sequencing results and *APPAT* expression in VSMCs.** (A) Venn diagram showed the total 2,037 candidate lncRNAs after coding potential analyses. (B) Validation of relative expression of four selected lncRNAs in ox-LDL-treated rabbit VSMCs and control. The differential expression results was consistent with the RNA-seq data ($n = 3$, * $P < 0.05$, ** $P < 0.01$, *** $P < 0.001$ vs control group). (C) Significant down-regulation of *APPAT* was detected in ox-LDL treated human VSMCs ($n = 4$, ** $P < 0.01$, *** $P < 0.001$ vs control group). * Student $t$-test was used in compression between two groups.

## Selection and validation of lncRNAs from rabbit and human

Two up-regulated (TCONS_02288701, TCONS_02225105) and two down-regulated (TCONS_00489746, TCONS_02443383) lncRNAs were selected for validation of RNA sequencing data using qPCR. All of these selected lncRNA transcripts were successfully amplified using designed primers (Table S2) and exhibited statistically differential expression between ox-LDL-treated and control groups ($n = 3$, $P < 0.01$) (Fig. 1B). The expression pattern of these four lncRNA were consistent with their RNA sequencing data.

These four selected rabbit lncRNAs sequences were chosen and compared with human sequence data on the NCBI website. Within the blast result, human lncRNAs with relatively high alignment identity were selected and amplified from total RNA production of human VSMCs using designed primer respectively (Table S3). Finally, ENST00000620272, corresponding to rabbit lncRNA transcript TCONS_00489746, was successfully amplified from human VSMCs and prepared for the subsequent study. ENST00000620272, renamed as *APPAT*, was a long intergenic non-protein coding RNA, which firstly generated from sequencing data of human blood sample. The protein-coding potential of *APPAT* was assessed using PhyloCSF and gained a score <20 which met the threshold as lncRNA (*Lin, Jungreis & Kellis, 2011*; *Pauli et al., 2012*).

The expression level of *APPAT* was checked by qPCR. A similar expressing trend was found between *APPAT* and TCONS_00489746. The former showed significantly decrease ($n = 4$, $P < 0.01$) in ox-LDL treated group compared with control group in human VSMCs (Fig. 1C).

## Distribution and location of *APPAT* in tissue and subcellular level

Since VSMCs was the major constituent of tunica media of artery, and *APPAT* could be successfully amplified from cultured VSMCs, samples of coronary artery was collected to determine the distribution and subcellular location of *APPAT*. We utilized designed

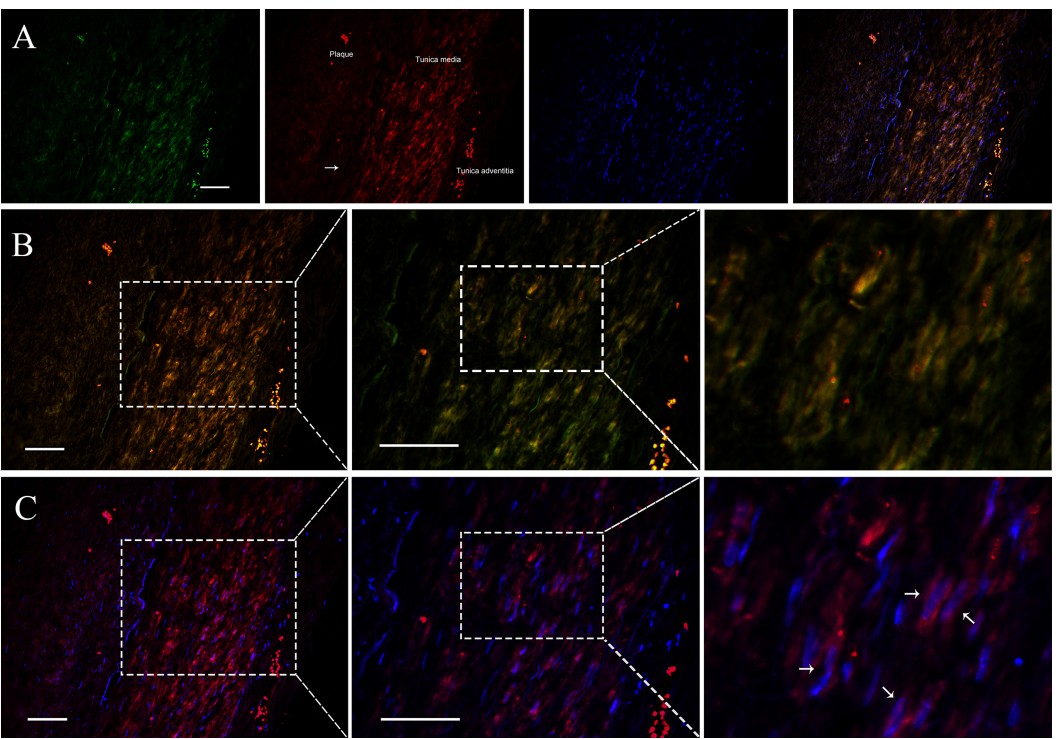

**Figure 2** *APPAT* **distribution and subcellular location.** IF-FISH was performed on coronary artery sample. DAPI, blue; *APPAT*, red; α-SMA, green. Scale bar, 50 μm. (A) Tunica media, tunica adventitia and plaque area were labeled on red fluorescent image, and white arrow showed the boundary of tunica intima. (B) Images of red and green fluorescent signals were merged to detect the co-expression of *AP-PAT* and *α*-SMA (mainly located in tunica media). (C) Red and blue fluorescent signals were merged to explore the subcellular distribution of *APPAT*. White arrows pointed to the merged area of DAPI (nuclear area) and *APPAT* (cytoplasmic area).

5'Cy3-probe for RNA IF-FISH to investigate visuospatial information of *APPAT* within human coronary artery (Table S4). On the IF-FISH images of coronary artery slides, *APPAT* were intriguingly located in VSMCs of tunica media of coronary artery but there was no obvious stain in tunica intima and adventitia (Figs. 2A and 2B). By contrast, the atherosclerotic plaque area on slide exhibited negative fluorescent signals of *APPAT* probe as the α-SMA probe did (Fig. 2B). *APPAT* and α-SMA exhibited similarly distributing and co-expression pattern. Furthermore, it could be distinguished under high magnification that *APPAT* was mainly localized in the cytoplasm but not the nuclear area (Fig. 2C).

### *APPAT* level in coronary artery sample with severe stenosis

Level of coronary artery stenosis was utilized to reflect the development of atherosclerosis plaque. The normal group contained coronary artery samples of dead people under unambiguously determined cause of death. Those samples with no obvious plaque were selected as the normal group (Fig. 3A). Samples of coronary artery with III-IV stenosis (Fig. 3B) were selected as pathology group after forensic pathological diagnosis (*Chen & Zhou, 2015*).

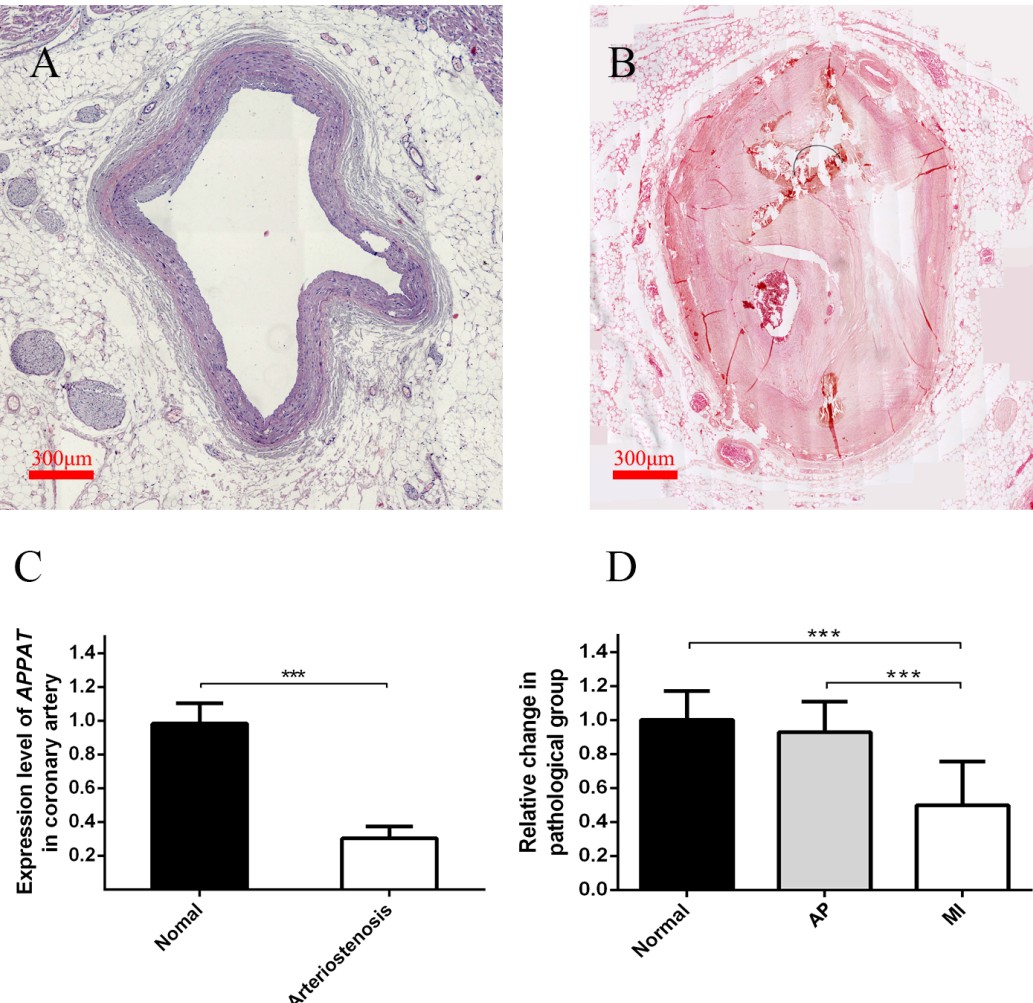

**Figure 3  Expression level of circulating *APPAT* in coronary artery tissue and blood samples.** (A) Normal coronary artery sample. (B) Coronary artery with severe arteriostenosis. (C) Expression level of *APPAT* were quantified by qPCR in normal ($n = 21$) and arteriostenosis ($n = 28$). Significant difference were detected between two groups (*** $P < 0.001$ vs normal). (D) Expression level of circulating *APPAT* in plasma of patients with angina pectoris ($n = 48$), MI ($n = 47$) and healthy group ($n = 43$). A decreasing tendency was detected from healthy to MI (*** $P < 0.001$ vs healthy group).

A statistically significant down-regulation was detected in the arteriostenosis group ($P < 0.01$) (Fig. 3C). The sharp decline of *APPAT* in arteriostenosis group seemed identical with the severity of pathological artery stenosis.

## Detection of *APPAT* in circulating blood of patients with coronary artery disease

For patients characteristics, see Table 1. The level of *APPAT* in extracellular circulating blood was investigated for whether it was detectable in blood sample, as well as existence of any dysregulation in atherosclerosis pathophysiologic process. Total RNA production was

**Table 1 Patient characteristics.**

| Characteristics | Normal ($n = 43$) | Angina pectoris ($n = 48$) | P | MI ($n = 47$) | P |
|---|---|---|---|---|---|
| Age (years) | $63 \pm 4.84$ | $65 \pm 6.62$ | 0.08 | $63 \pm 8.25$ | 0.75 |
| Male/Female (n/n) | 24/19 | 22/26 | – | 26/21 | – |
| HDL (mmol/L) | $1.06 \pm 0.39$ | $1.10 \pm 0.34$ | 0.64 | $1.04 \pm 0.23$ | 0.58 |
| LDL (mmol/L) | $2.04 \pm 0.79$ | $2.07 \pm 0.61$ | 0.81 | $1.91 \pm 0.98$ | 0.47 |
| cTnI (ng/mL) | $0.085 \pm 0.070$ | $0.176 \pm 0.192$ | <0.01 | $3.962 \pm 4.299$ | <0.01 |
| FBG (mmol/L) | $5.69 \pm 2.07$ | $5.80 \pm 0.83$ | 0.74 | $7.05 \pm 4.31$ | 0.81 |
| CK-MB (U/L) | $24.5 \pm 7.78$ | $17.73 \pm 5.93$ | <0.01 | $27.58 \pm 12.55$ | <0.01 |
| Diabetes (n, %) | 5, 11.63 | 11, 22.96 | – | 24, 51.06 | – |
| Hypertension (n, %) | 13, 30.23 | 20, 41.67 | – | 34, 72.34 | – |

Notes.

HDL, high density lipoprotein; LDL, low density lipoprotein; cTnI, cardiac troponin I; FBG, fasting blood glucose; CK-MB, creatinine kinase-MB.

extracted and successfully amplified in all three groups. An obviously decreasing tendency of circulating *APPAT* was found from normal, to AP group, then to MI group (Fig. 3D). The level of *APPAT* was slightly decreased in AP group without statistical significance, but sharply declined in the MI group. This result exhibited the down-regulation of *APPAT* was identical with the pathological progress of MI, since AP was usually treated as the early stage of coronary heart disease ($P < 0.01$).

## The effect of diabetes and hypertension on *APPAT* level in circulating blood

The clinical diagnosis of hypertension and diabetes of those patients with MI were investigated. Patient with MI was further classified as MI (none of hypertension or diabetes) group with MI+hypertension group and MI+diabetes group respectively. No significant differences of *APPAT* level was found between MI ($n = 23$) and MI+hypertension group ($n = 24$) ($P > 0.05$) (Fig. S1A), and a similar situation found in MI ($n = 13$) and Mi+diabetes group ($n = 34$) ($P > 0.05$) (Fig. S1B).

## Predictive power of circulating *APPAT* for patients with AP or MI

The ROC curve displayed the relationship regarding sensitivity and specificity for the *APPAT* to predict the patients with AP or MI. In addition, ROC curve of cTnI level in blood was calculated and compared with that of *APPAT*. A similar predictive power between cTnI (sensitivity: 80.85, specificity: 97.67) and *APPAT* (sensitivity: 78.72, specificity: 93.02) was found. ROC analysis of *APPAT* revealed an AUC of 0.9302 (95% confidence interval CI [0.8795–0.9810], $P < 0.0001$) (Fig. 4A) in MI group, whereas AUC of cTnI was 0.8951 (95% CI [0.8247–0.9655], $P < 0.0001$). Comparison of AUCs between *APPAT* and cTnI was calculated, and there was no significant difference ($P = 0.4678$).

By contrast, ROCs curves revealed low predicting power of *APPAT* (sensitivity: 89.58, specificity: 30.23) and cTnI (sensitivity: 45.83, specificity: 90.7) in AP patients, and relatively low AUC was also found both in *APPAT* (0.5833 (95% CI [0.4641–0.7026]), $P = 0.1716$) and cTnI (0.6221 (95% CI [0.5027–0.7415]), $P = 0.04520$) (Fig. 4B). The AUCs comparison was calculated also ($P = 0.6420$).

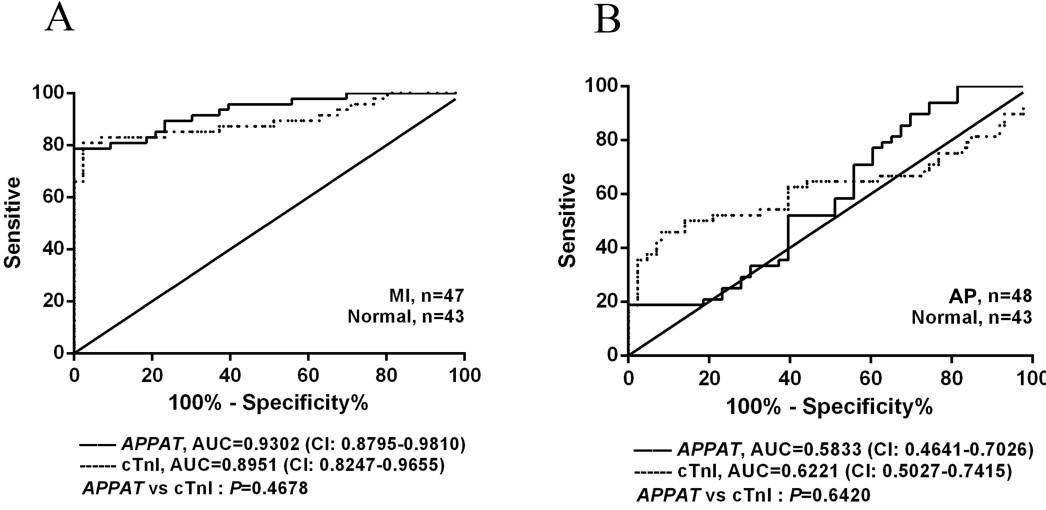

**Figure 4** ROC curve analyses of *APPAT* and cTnI in angina pectoris and MI patients. AUC indicates area under the ROC curve. Comparison between *APPAT* and cTnI in MI patients (A) and angina pectoris patients (B). No significant difference was found between *APPAT* and cTnI in MI or angina pectoris predicting ability.

## Prediction on miRNA targets for *APPAT* and primary validation

Cytoplasmic lncRNAs could act as a sponge for miRNAs and thus play role in a competing endogenous RNAs (ceRNAs) regulation loop, which derepressing associated mRNAs by decreasing concentration of targeting microRNA (miRNA) (*Chen, 2016*). We applied the pictar, miRDB, lncBase and mirna22 online databases to search for the potential miRNAs for *APPAT*, and calculated the intersection of these predicting results assemblages. Then candidate miRNAs were primarily selected as which were at least included in three databases (Fig. S2). Further screening work was based on an integrated consideration of hybrid ability, number of biding site, sequence identity, even the research background of them. Three miRNAs, miR-135b, miR-647 and miR-1229 were selected as candidate targets of *APPAT* (Table S5). All of those three miRNAs could be amplified from RNA extraction of human VSMCs. Moreover, we validated the expressing change of miRNAs in human VSMCs after ox-LDL treatment. The results exhibited miR-647 was significantly up-regulated in ox-LDL treated VSMCs, whereas no obvious change in miR-135b and miR-1229 ($n = 3$, $P < 0.05$) (Fig. 5A). We selected miR-647 as the potential miRNA target of *APPAT* for the further analyses.

## MiR-647 level in coronary artery and blood samples

The expression level of miR-647 in samples with stenosis ($n = 28$) was almost three folds than that of normal group ($n = 21$) (Fig. 5B). Furthermore, a statistically significant up-regulation of circulating miR-647 was found in the MI group ($n = 47$) compared with the normal subjects ($n = 43$) (Fig. 5C). The altering tendency was coincident with the founding of coronary artery sample.

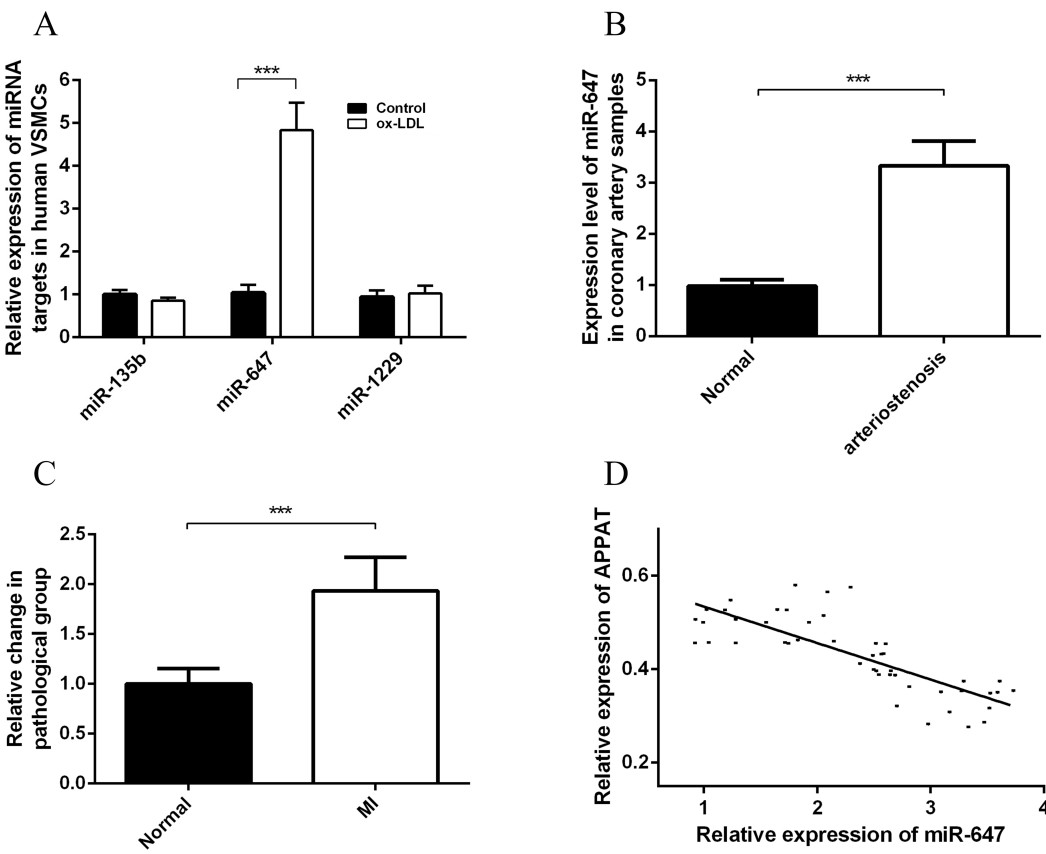

**Figure 5  Expression status of miR-647 and correlation analyses with *APPAT*.** (A) Expression levels of miRNA targets of *APPAT* in ox-LDL treated human VSMCs and control. (B) The expression level of miR-647 in normal ($n = 21$) and arteriostenosis ($n = 28$) samples. (C) Expression levels of circulating miR-647 in plasma of patients with MI ($n = 47$) and healthy group ($n = 43$). (D) Correlation analysis of expression between circulating *APPAT* and circulating miR-647 in MI patients ($n = 47$, $r = -0.7908$, $P < 0.0001$).

The correlation analyses between level of circulating miR-647 and circulating *APPAT* in blood samples of myocardial infraction group was calculated. The level of circulating *APPAT* was negatively correlated with that of circulating miR-647($r = -0.7908$, $P < 0.01$) (Fig. 5D).

## DISCUSSION

Increasing evidences shows that lncRNAs with various classification plays the critical role in atherosclerosis, although they were considered as junk of genome in the past (*Holdt et al., 2013*; *Michalik et al., 2014*). LncRNAs generated from different cell types in vascular wall implicated in various aspects of disease progression, like vascular function, metabolic regulation, inflammation and immunity (*Carpenter et al., 2013*; *Cui et al., 2015*; *Hu et al., 2015*; *Shan et al., 2016*). While the majority of studies have focused on the intracellular roles of lncRNAs, there is increasing interest in the potential roles of extracellular circulating lncRNAs. The release of cellular RNA species into circulation may reflect disease-specific

tissue injury and/or remodeling, or potential intercellular signaling. The intergenic lncRNA-*APPAT* investigated in present paper could be amplified from human VSMCs and alteration of expressing level could be induced under stimulation of ox-LDL *in vitro*. Moreover, the expressions of *APPAT* and its targeting miR-647 were detectable in circulating blood and pathological tissue both. The dynamical alteration of *APPAT* might provide the potential value on the development of disease in clinical diagnosis.

As a main component of vessel wall, VSMCs performed vital function during the progression of atherosclerosis in response to various risk factors within plaques (*Bennett, Sinha & Owens, 2016*; *Chistiakov, Orekhov & Bobryshev, 2015*). The proliferation or apoptosis in VSMCs may result in atherosclerosis plaque instability and rupture (*Chen et al., 2004*; *Ding et al., 2012*; *Okura et al., 2000*). In this study, the expression of *APPAT* was significantly repressed under ox-LDL stimulation in human VSMCs. This reduction might associated with the progression of VSMCs proliferation or apoptosis, which needs further investigation on its target mRNAs and the molecular mechanism.

It is becoming clear that many lncRNAs express in specific cell type or tissues, and function of lncRNAs was associated with their unique subcellular localization as nuclear enriched, cytoplasmic enriched, or both (*Batista & Chang, 2013*; *Chen, 2016*; *Chen & Carmichael, 2010*). The *APPAT* restrictively distributed in tunica media of artery, where mainly consisted of VMSCs. In the development of atherosclerosis plaque, the media oriented VSMCs differentiate from contractile to synthetic phenotype, and accumulate into the intima by proliferation or directed migration, which play a vital role in this pathogenesis progress. Together with the negative fluorescent signals in adventitia and plaque area, it is suggested that *APPAT* might highly express in contractile phenotype of VSMCs other than synthetic phenotype, since the latter commonly migrated into plaque area. Similar distribution pattern was detected from signal of α-SMA probe which was designed for contractile VSMCs stain specially. Furthermore, subcellular location of *APPAT* showed that it possibly performs the function in cytoplasm which mean it may plays regulatory roles on post-transcription. Unlike mitochondrial genome originated lncRNAs which were mainly expressed in heart (*Kumarswamy et al., 2014*), *APPAT* gene was localized on autosomal (chromosomes 2) and should initially generated in nucleus but functioned in cytoplasm. Here we inferred *APPAT* was a contractile VSMCs enriched cytoplasmic lncRNA.

The alteration of lncRNA level in cell may somewhat reflected on its total expression amount in tissue level. A meaningful change should be detectable when pathological symptom happened. We further confirmed this assumption that the expression level of *APPAT* was significantly declined in coronary artery tissue with serious stenosis. This result suggested that the expression level of *APPAT* seemed closely related with the severity of arteriostenosis which indicated by the pathological development of atherosclerotic plaques. Similarly, *lincRNA-p21* exhibited obviously lower expression in coronary artery samples of vascular disease patients compared to control subjects. It was believed *lincRNA-p21* functioned in the development of atherosclerosis and coronary artery disease (*Wu et al., 2014*). In this report, it has demonstrated that ox-LDL stimulation could reduce *APPAT*, and *APPAT* enriched in contractile VSMCs of tunica media. Together with the down-regulation in coronary artery of CAD patient, we inferred that *APPAT* possibly plays

key role in maintaining phenotype of VSMCs in disease progression, thus influencing the severity of coronary artery stenosis, especially the vulnerable plaque stability.

Ideal biomarkers for noninvasive diagnosis should be stable and easily detected with high sensitivity and specificity (*Qi, Zhou & Du, 2016*; *Shi & Yang, 2016*). Few evidence of utilizing circulating lncRNAs in diagnosing/prognosing cardiovascular disease has been reported, and it was encouraged that such experiment should be developed on human cells and clinical samples preferably (*Beltrami, Angelini & Emanueli, 2015*). The mitochondrial long noncoding RNA-*LIPCAR* was found with a potential value of predicting future death in patients with heart failure. It showed a biphasic altering direction in which lncRNA-*LIPCAR* down-regulated after MI happened but up-regulated in subsequence stage (*Kumarswamy et al., 2014*). Circulating long noncoding RNA *UCA1* was reported as a novel biomarker of acute MI recently (*Yan et al., 2016*). As well as, CoroMarker was also uncovered from circulating peripheral blood and possibly involved in the pathogenesis of coronary artery disease (*Yang et al., 2015*). In present study, a clearly down-regulating tendency of *APPAT* could be found from health group to AP group, then to the MI patients. It was reasonable to think this expressing tendency in circulating blood system may in line with the simultaneously pathological dysregulation of *APPAT* in coronary artery tissue. Furthermore, this down regulation seemed independent from the existence of hypertension and diabetes. The ROC analysis exhibited a similar tendency to that of cTnI. The dynamic change of *APPAT* might contribute to the indication of atherosclerotic plaque development before severe cardiovascular symptom happen. A more precisely accession on the predicting power of *APPAT* called for a relatively lager samples size which should be carried out in the future study.

Cytoplasmic located LncRNAs could influent miRNAs function and associated mRNAs expression by acting as competing endogenous RNA (ceRNA), which was described as a feed-forward intracellular regulatory loop (*Cesana et al., 2011*; *Rashid, Shah & Shan, 2016*; *Wang et al., 2010*). The ceRNAs indirectly correlate with each other by competing for miRNAs containing same binding site, thus regulate the availability of miRNAs (*Tay, Rinn & Pandolfi, 2014*). MiR-647 was a newly found biomarker for gastric cancer (*Rawlings-Goss, Campbell & Tishkoff, 2014*; *Yang et al., 2014*). Evidence from limited studies suggested that overexpression of miR-647 in cancer cell *in vitro* could inhibit cell proliferation and increase cell apoptosis (*Cao et al., 2017*). In the present study, we found an increasing tendency of miR-647 expression in ox-LDL treated VSMCs, pathological coronary artery samples and circulating blood samples of patients with cardiovascular disease, which was obviously opposite to that of *APPAT*. Correlation analysis exhibited a negative correlation between miR-647 and *APPAT* further. This opposite expressing tendency between lncRNA and its targeting miRNA might be attributed to the ceRNA regulation, and finally resulted in corresponding change of their expressions at tissue and circulating blood level. Disease-specific up- or down-regulation of individual lncRNAs with the following alteration in ceRNA-mediated interactions may cause great effects in pathophysiological conditions (*Tay, Rinn & Pandolfi, 2014*). This sort of co-expression may assist in understanding the function and role of *APPAT* in the pathogenesis of disease based on examination of blood samples.

## CONCLUSIONS

Based on the aforementioned, we assumed that circulating lncRNA-*APPAT* may perform an important function and have some indicating ability on the development of atherosclerosis. Further research will put effort into exploring the mechanism by which *APPAT* plays function in the pathological progression of atherosclerotic disease.

**List of abbreviations**

| | |
|---|---|
| **VSMCs** | vascular smooth muscle cells |
| **ox-LDL** | oxidized low density lipoprotein |
| **FFPE** | formalin-fixed and paraffin-embedded |
| **lncRNA** | long noncoding RNA |
| **qPCR** | quantitative real-time PCR |
| **miRNA** | microRNA |
| **ceRNA** | competing endogenous RNA |
| **MI** | myocardial infarction |
| **AP** | angina pectoris |
| **CNCI** | coding-non-coding index |
| **CPC** | coding potential calculator |
| **PFAM** | Pfam-scan |
| **phyloCSF** | phylogenetic codon substitution frequencies |

### Funding

This work was supported by the National Natural Science Foundation of China [grant number 81270371, 81770462], and the Science and Technology Program of Hunan Province (2015RS4014). The funders had no role in study design, data collection and analysis, decision to publish, or preparation of the manuscript.

### Grant Disclosures

The following grant information was disclosed by the authors:
National Natural Science Foundation of China: 81270371, 81770462.
The Science and Technology Program of Hunan Province: 2015RS4014.

### Competing Interests

The authors declare there are no competing interests.

### Author Contributions

- Fanming Meng conceived and designed the experiments, performed the experiments, analyzed the data, wrote the paper, prepared figures and/or tables, reviewed drafts of the paper.
- Jie Yan, Yunjuan Jiao and Fan Yang contributed reagents/materials/analysis tools.
- Qiongshan Ma reviewed drafts of the paper.
- Luyang Han performed the experiments.

- Jing Xu analyzed the data.
- Junwen Liu conceived and designed the experiments, wrote the paper, reviewed drafts of the paper.

## Human Ethics

The following information was supplied relating to ethical approvals (i.e., approving body and any reference numbers):

The Institutional Ethics Committee (The Third Xiang Ya Hospital of Central South University) approved the study and written informed consent was obtained from all patients (approval number: 2015-S177).

## Data Availability

Meng, Fanming (2017): RNA-seq data of ox-LDL treated rabbit VSMCs. figshare. https://doi.org/10.6084/m9.figshare.5446705.v1.

Meng, Fanming (2017): raw data and information of human participants. figshare. https://doi.org/10.6084/m9.figshare.5480389.v1.

## Supplemental Information

Supplemental information for this article can be found online at http://dx.doi.org/10.7717/peerj.4246#supplemental-information.

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
