# Peer review of "Expression status and clinical significance of lncRNA APPAT in the progression of atherosclerosis"

_PeerJ, doi:10.7717/peerj.4246_

## Round 0.1 · original submission · Major Revisions

I have reviewed the manuscript and the reviewer comments and based on that I certainly believe this manuscript has novelty and interest enough for a publication. However some of the reviewers raised some important points that I would request you to address. Particularly I think the suggestion to share the data e.g. in the SRA is an important point.

·

Basic reporting

Meng et al have conducted a study of APPAT lncRNA and its use as a biomarker in circulating blood samples of angina and MI patients. The study seems to have an impressive depth and is overall well-conducted. However it suffers from lack of structure and stringency in reporting and could be much improved. While the English language itself is reasonable, I believe that the manuscript would benefit a lot from a work-through by a key-author, at each section asking if 1) the key points are clearly communicated, and 2) if superfluous information not necessary for key points could be removed. I have tried to give some examples below:
1. One overall structure-suggestion. Shorten materiel and method (5 pages now)
It would be much better that each result section match med each figure.
2. How to make the ox-LDL treated VSMCs of rabbit? What is the treatment? How? What is control for the treated cells?
3. ‘Randomly’ selected 2 up and 2 down regulated ? – I believe the word ‘randomly’ can be better described. It is important for the structure of the article.
4. Figure 1B, there are no significant stars. How to calculate the significance? How many times was the experiment repeated? This should clearly be shown, also in caption
5. Short description of results in figure captions.
6. In figure 2, co-expression of APPAT and α-SMA is hard to see. There should be better indication of two kinds of scale bar.
7. How to evaluate the expression level in fig 3?? There is no any description about it.

Experimental design

adressed in previous box

Validity of the findings

adressed in previous box (but seems valid)

·

Basic reporting

The authors have done a fine work in terms of professional English, providing background and describing the need/caveat the address.

Some points to address:
In the abstract, at lines 32-to-38, it states "To determine expression alteration of APPAT in coronary artery, we performed qPCR on samples of healthy and pathological coronary artery. Same test of APPAT level was performed on circulating blood samples from the normal and patients with angina pectoris or myocardial infarction, followed by calculation of prediction power of APPAT. Additionally, we predicted microRNA target of APPAT and the expression level of the target was primarily measured in human VSMCs cell line, coronary artery, and blood samples."
To me this doesn't read well. Something like "...artery. We also assessed circulating APPAT levels in blood samples from healthy individuals, and patients with angina pectoris (AP) or myocardial infarction (MI). Additionally, we predicted microRNA targets of APPAT, and show that the expression level of the targets was primarily measured in human VSMCs cell line, coronary artery, and blood samples. Lastly, we examined the predictive power of APPAT for the risk of AP or MI." would perhaps be better.

Experimental design

A power analysis of
- discovery experiment
- predictive power (ROC and AUC)
is lacking. Please add this.

Also include an assessment of the validity of the discovery given the many potential non-coding and coding RNAs that exist. In other words, is p = 0.05 a valid threshold? This threshold of discovery should be corrected for multiple testing.

Validity of the findings

Figure 1A:
I am missing the explanations for the abbreviations in the figure. What does CPC, PFAM, etc stand for?

Figure 1B: Perhaps I missed that in the manuscript, how do you explain the relative lower expression of APPAT in human cells as compared to rabbit cells after treatment with oxLDL?

Figure 2: Please include non-fluorescent HE, EvG and alpha-SMC stained histological slides of the same artery/plaques. This will aid in orientation with respect to the lumen and the remaining non-plaque area as compared to the IF-FISH images. This will also mean that Figure 3 can be dropped, unless it is the same sample. But together these Figures are not informative at the moment.

Replication & Figure 4A:
- Please include data on the comparison between the two AUCs for APPAT and troponin.
- Is the AUC of APPAT *significantly* better then the one for troponin in predicting MI? In other words: what is the added value? Why not just use troponin?
- This is also lacking in lines 274-282: what is the statistical significance of these ROC/AUC analyses?
- Please, also include a power analysis of the ROC/AUC analysis.
- Furthermore, what is lacking is a replication of these results in a (larger) independent study. This is absolutely necessary and of great value to understand the relevance of APPAT as compared to for instance troponin as well as the validity of APPAT in predicting disease outcome.
- This has also implications for the discussion at lines 366-370 regarding the potential value of APPAT as a prognostic marker. I don't believe there can even be a discussion about this, as the validity of APPAT has not been tested in independent studies. This should be addressed in experiments and in the discussion.

The above also has implications for the conclusions made in lines 390-392. Things that would aid in support of the conclusions, and that would aid in assessing the validity of APPAT and miRNA-647:
- Please add in data on correlations (of APPAT and miRNA-647) in larger datasets with sub-phenotypes of atherosclerosis, for instance coronary artery calcification, and carotid IMT as these are strongly correlated to cardiovascular outcome.
- Are APPAT and the miRNA-647 correlated to other cardiovascular risk factors? One would assume there should be correlations with CAC, cIMT and risk factors, if APPAT and miRNA-647 are valid candidates diagnosis and prognosis.
- Are APPAT and miRNA-647 also present in carotid plaques? Or is it merely constraint to coronary arteries?
- Are APPAT and miRNA-647 also expressed in biosamples (i.e arteries) from GTEx Portal?

Figure 5: In figure 1B 'treated' is noted as 'oxLDL', for consistency that should also be the case for figure 5A.

Additional comments

In general nicely written, and well structured. What is really necessary though with this type of discovery efforts is to assess the validity of novel targets (verify) and replicate findings in independent and larger samples.

Reviewer 3 ·

Basic reporting

The authors propose the lncRNA APPAT as new player in the progression of atherosclerosis. The work is very interesting and potentially useful in future clinical studies. English could be improved in some parts of the manuscript.

The authors should share the raw data in public archives such as SRA or GEO before publication. It's possible to submit the data in a private format and then release them after publication. An accession number has to be provided in the manuscript, in the section regarding RNA-seq (methods section).

The section regarding miRNA target prediction is not completely clear. How did you select the miRNA candidates? Did you performed an intersection of target prediction algorithms? From the text this part is not clear. I suggest to write a paragraph about that in the methods.

Another point to be clarified is the p-value. Did you adjust the p-value according to FDR or other approaches? From the text it seems you used only the p-value in all the analyses.

Experimental design

The experimental design is correct and the use of NGS improves the quality of the results compared to the use of RT-PCR.

As suggested by the authors at the end of the paper, it is interesting to explore the mechanism of action of APPAT and also the potential interactions with miRNAs. I suggest to focus on extracellular vesicles extracted from serum and plasma for future experiments. In this way, you could be able to understand if APPAT is actively released in the bloodstream or not, and if the content of these vesicles is released inside target cells. Consequently, you could identify the "sponge" effect of APPAT in the target cell.

Validity of the findings

The findings support the hypothesis but future experiments are needed in order to validate the role of APPAT as non-invasive biomarker and its function and role in the progression of atherosclerosis.

---

## Round 0.2 · accepted · Accept

I have gone through your rebuttal and also had one of the reviewers take another look at it, and we both agree that the manuscript is now ready for publication.

·

Basic reporting

The authors made good efforts to improve the quality of the article. There are no further questions from me. I hope that the authors will give more contributions in the field of atherosclerosis pathogenesis in the near future.

Experimental design

-

Validity of the findings

-

Additional comments

The authors made good efforts to improve the quality of the article. There are no further questions from me. I hope that the authors will give more contributions in the field of atherosclerosis pathogenesis in the near future.